

# Virtual mapping and analytical data integration: A teaching module using Precambrian crystalline basement in Colorado's Front Range (USA)

Kevin H. Mahan[1], Michael G. Frothingham[1], and Ellen Alexander[1]

[1]Department of Geological Sciences, University of Colorado Boulder, 2200 Colorado Ave., Boulder, Colorado 80309

**Correspondence:** Kevin H. Mahan (kevin.mahan@colorado.edu)

**Abstract.** The COVID-19 pandemic hindered the ability to conduct field geology courses in a hands-on and boots-on traditional manner. In response, we designed a multi-part virtual field module that encompasses many of the basic requirements of an advanced field exercise, including designing a mapping strategy, collecting and processing field observations, synthesizing data from field-based and laboratory analyses, and communicating the results to a broad audience. For the mapping exercise, which is set in deformed Proterozoic crystalline basement exposed in the Front Range of Colorado (USA), student groups make daily navigational decisions and choose stations based on topographic maps, Google Earth satellite imagery, and iterative geological reasoning. For each station, students receive outcrop descriptions, measurements, and photographs from which they input field data and create geologic maps using StraboSpot. Building on the mapping exercise, student groups then choose from six supplements, including advanced field structure, microstructure, metamorphic petrology, and several geochronological datasets. Because scientific projects rarely end when the mapping is complete, the students are challenged to see how samples and analytical data may commonly be collected and integrated with field observations to produce a more holistic understanding of the geological history of the field area. While a virtual course cannot replace the actual field experience, modules like the one shared here can successfully address, or even improve on, some of the key learning objectives that are common to field-based capstone experiences, while also fostering a more accessible and inclusive learning environment for all students.

## 1  Introduction

The Covid-19 pandemic hindered our ability to conduct in-person field geology courses, and prompted world-wide efforts to design effective alternative on-line educational experiences. We designed an activity that can be delivered remotely and conducted virtually while still providing an effective learning experience centered on field mapping skills. We also wanted to create a capstone experience that challenges students to go beyond creating a map and understanding the basic 3-dimensionality of geologic structures, but also to gain a deeper appreciation for how scientific endeavors are commonly conducted through subsequent laboratory analyses and integration of the results with field-based relationships.

Field mapping exercises have traditionally been a central component of undergraduate geology curricula. Historically, development of field geology skills was integral to students' preparation for entry into the geoscience workforce (Heath, 2003; Whitmeyer et al., 2009b). Recent educational research has additionally identified benefits of field work across multiple disci-





plines, including improved student motivation and learning outcomes relative to traditional classroom-based education (Stokes and Boyle, 2009; Fedesco et al., 2020). The 2021 report from the Future of Undergraduate Geoscience Education initiative (Mosher and Keane, 2021) found that collaborative, problem-based learning exercises—typical of both virtual and in-person field activities—not only improve overall learning outcomes, but also are specifically beneficial for inclusion and retention of students from historically underrepresented minority groups. Virtual field exercises provide similar educational benefits

with enhanced accessibility (Stokes et al., 2019) and decreased cost of participation relative to in-person field work (Abeyta et al., 2021). Furthermore, with rapid improvement and evolution in digital mapping and online collaboration tools, integrating modern technology into education is a valuable addition to virtual and traditional in-person field courses (Whitmeyer et al., 2009a).

In this contribution, we describe a multi-part activity involving virtual mapping and group collaboration with associated

analytical datasets. The first part is a mapping exercise that simulates doing field work in the Front Range of Colorado (USA) while creating a geologic map and cross-section of deformed igneous and metamorphic rocks (Fig. 1). The second part introduces additional analytical datasets (e.g., microstructure, petrology, and geochronology) from the same field area, which students incorporate into their study of the area's geological history. The activity is intended for an upper-level undergraduate field course for geology majors. Students should have already been introduced to basic field methods and an introductory struc-

tural geology course. Ideally, students will have also had at least one of the following additional courses of introductory earth materials, mineralogy, petrology, or geochemistry, but the activities can be managed in such a way that those pre-requisites may not be necessary. The module should take approximately 2 weeks to complete in an immersive field course (all day, every day), or could be spread over a longer period during a semester.

**Table 1.** Learning Objectives

| No. | Learning Objective |
| --- | --- |
| 1: | Design a field strategy to collect or select data in order to answer a geologic question |
| 2: | Collect accurate and sufficient data on field relationships and record these using disciplinary conventions (field notes, map symbols, etc.) |
| 3: | Synthesize geologic data and integrate with core concepts and skills into a cohesive spatial and temporal scientific interpretation |
| 4: | Interpret earth systems and past/current/future processes using multiple lines of spatially distributed evidence |
| 5: | Develop an argument that is consistent with available evidence and uncertainty |
| 6: | Communicate clearly using written, verbal, and/or visual media (e.g., maps, cross-sections, reports) with discipline-specific terminology appropriate to your audience |
| 7: | Work effectively independently and collaboratively (e.g., commitment, reliability, leadership, open for advice, channels of communication, supportive, inclusive) |

The Designing Remote Field Experiences project was sponsored by the National Association for Geoscience Teachers and International Association for Geoscience Diversity and the National Science Foundation.



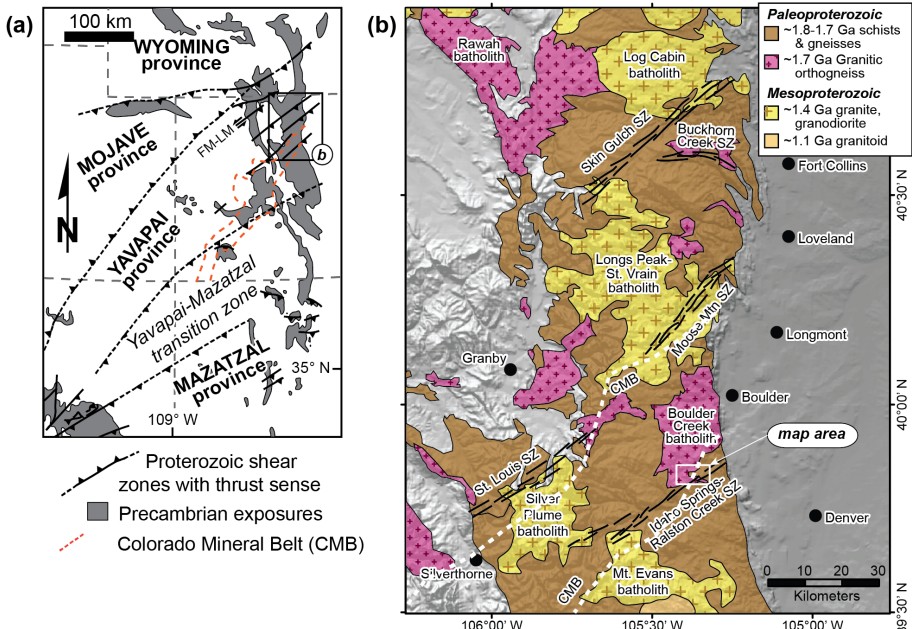

**Figure 1.** (a) Map of exposed Proterozoic rocks in southwestern United States with major provinces and shear zones (modified after Karlstrom and Williams (2006)). FM-LM: Farwell Mountain-Lester Mountain zone. (b) Simplified geological map of Proterozoic exposures in the Front Range (modified after Tweto (1979)). Colorado Mineral Belt boundaries in both (a) and (b) approximated from Tweto and Sims (1963), Chapin (2012), and Caine et al. (2010).

## 2 Framework - Learning objectives, Materials provided, Technology requirements

### 2.1 Learning objectives

Participants in the 2020 Designing Remote Field Experiences project (Atchison et al., 2021) developed a flexible set of learning objectives to which most capstone field courses can target (whether delivered remotely or in-person). This module is designed to address a subset of those objectives (Table 1). Here, we briefly describe the mechanisms for each objective.

For Objective 1, student groups (e.g., mapping partners) select a fixed number of stations for each of the "field mapping days" and justify their requests based on considerations of safety, access, exposure, and geological reasoning. For Objective 2, students collect map data via "virtual outcrops" that include information such as rock types, features, fabrics, structures, measurements, photos, and sketches for each selected station. Objectives 3 and 4 are addressed through interpreting map relations using fundamental geological principles during each virtual mapping day, as well as integrating the results from additional dataset analysis during Part II. For Objective 5, students develop multiple working hypotheses throughout their mapping, design data acquisition strategies to efficiently test those hypotheses, identify uncertainty from input virtual outcrop data, and project this uncertainty (e.g., certain vs approximate contacts) into their preferred mapping interpretations that best fit the available evidence. The combination of outcrop interpretation and laboratory analytical data requires students to grapple





with a range of types of uncertainty, some of which is quantifiable in a straightforward manner and some of which is not. For Objective 6, this module strengthens modern communication skills by encouraging the use of video conferencing tools with screen-sharing for collaborative work, interactive learning, and professional presentations; each of which pair well with digital technology for mapping and quantitative analysis. Addressing Objective 7, students participate in this module through a combination of whole-class discussions, individual assignments, small group collaborative mapping, small group drop-in meetings, and small group oral presentations to the entire class.

## 2.2 Materials provided

Module materials are listed in Table 2, described in the Appendix, and provided in supplemental materials. They can also be accessed from the Teach the Earth activities portal hosted by the Science Education Resource Center (SERC) at Carlton College (Mahan and Frothingham, 2021). For the Part I mapping exercise, the minimum required materials are the station descriptions (i.e., "virtual outcrops"). Additional resources include introductory materials, instructional documents, data request forms, station location shapefiles, basemaps, and other resources to aid instructors and students throughout the module (Appendix A1). For the Part II analytical data integration, students may have variable prior exposure to geochemistry, geochronology, petrology, and structural geology coursework. Therefore, a set of instructional resources was assembled to provide the requisite background knowledge to conduct such analyses. These resources are a combination of existing educational materials (textbook chapters, online lecture notes, and review articles) as well as handouts created with the relevant data sets in mind (Appendix A2). For Part III, an example set of instructional guidance and a grading rubric is also provided for the written report (Appendix A3).

## 2.3 Technology requirements and recommendations

During the Covid-19 pandemic, most students and instructors participated in this module from their homes using personal computers and internet. Thus, we designed the activities and materials such that all necessary software is either free or commonly accessible through university or college resources. Part I mapping uses Google Earth and Strabospot software (Walker et al., 2019). Both of these can be run as web, desktop, or mobile applications. We prefer to use the Google Earth Web project link because it seamlessly integrates station locations and oriented viewpoint photographs over satellite imagery and 3-d topography, without the need to download and install Google Earth Pro for desktop or associated datasets. The StraboSpot web application streamlines importing station shapefiles, interactive collaboration within mapping groups, and integration of digital mapping into final figures and deliverables. Alternatively, the StraboSpot mobile application also runs on tablets and mobile phones, and it provides additional offline mapping capabilities if desired. The detailed StraboSpot set-up instructions are optimized for the web version.

Part II analyses require a variety of web or desktop software, depending on the dataset and analytical technique. Most datasets require spreadsheet software (e.g., Microsoft Excel or Google Sheets), advanced structural analysis also requires free stereonet software such as Stereonet 11 (Cardozo and Allmendinger, 2013; Allmendinger et al., 2013) or Orient (Vollmer, 2015), and



**Table 2.** Supporting Materials

| Part I Mapping | Part II Analytical Datasets | Part III Reporting |
|---|---|---|
| Station descriptions and data | Introduction to datasets | Instructions for written report |
| Google Earth Web project | **Dataset 1** Advanced field analysis | |
| Station location shapefile | All field data spreadsheet | |
| Instructions for StraboSpot | **Dataset 2** Microstructural analysis | |
| Station request form | Quartz EBSD textfile | |
| Basemap template | **Dataset 3** Metamorphic petrology | |
| Map unit introductory slides | **Dataset 4** Monazite geochronology | |
| Assessment form | Monazite U-Th-total Pb spreadsheet | |
| Summary for Instructors | **Dataset 5** Igneous zircon geochronology | |
| | Igneous zircon U-Pb spreadsheet | |
| | **Dataset 6** Detrital zircon geochronology | |
| | Detrital zircon U-Pb spreadsheet | |
| | Geochronology Review | |
| | U-Pb Concordia spreadsheet | |

microstructural analysis requires either the freeware MTEX toolbox (Bachmann et al., 2010; Hielscher and Schaeben, 2008)
for MATLAB or the commercial Channel 5 software from Oxford Instruments.

## 3  Geologic Background

The field area is in the Front Range of Colorado (USA) and it hosts polydeformed intrusive and supracrustal crystalline rocks
(Fig. 1). The basic map-scale structures include a km-scale synform of Coal Creek quartzite and schist structurally overlying
Boulder Creek granodiorite, with one limb overprinted by the Idaho Springs-Ralston shear zone (Taylor, 1976; Widmann et al.,
2000; Kellogg et al., 2008). Outcrop-scale fabrics and structures include relict bedding, multiple generations of foliation and
schistosity, open to isoclinal folds with axial planar foliation, mylonitic foliation, mineral and stretching lineations, and locally
well-developed shear sense indicators.

The diverse geologic history of Colorado's Front Range, spanning ca. 1.8 billion years, offers several avenues to attract stu-
dents' interests and effectively motivate them to learn (e.g., Stokes and Boyle, 2009). Two potential perspectives on the region
may provide such motivation: 1) the features that illuminate how this part of Laurentia was built during Paleoproterozoic time
and 2) their possible relations to subsequent deformation and economic mineralization. First, the exposed rocks and structures
in this field area reflect the processes that formed Colorado's Proterozoic continental crust, including island-arc magmatism,
back-arc basin sedimentation and inversion, low to medium-pressure and high temperature regional metamorphism, and poly-
phase deformation (e.g., Whitmeyer and Karlstrom, 2007; Jones et al., 2009). We recommend providing students a short paper





to read at the beginning of this module to introduce some of these concepts in Colorado without directly addressing the targeted map area. The paired geological and seismic study of a terrane boundary in northernmost Colorado by Tyson et al. (2002) is a good example. Second, the map area lies along trend of the Colorado Mineral Belt (Fig. 1, which includes Cenozoic intrusions and associated mineral deposits (e.g., gold, silver, molybdenum) that have played a major role in Colorado's socioeconomic
development over the last 200 years (Chapin, 2012). Tweto and Sims (1963) suggested a Proterozoic ancestry to the belt based on a series of similarly oriented (NE-striking) ductile shear zones, one of which occurs in the map area (e.g., Idaho Springs-Ralston shear zone) and another of which is described by Tyson et al. (2002). Whether or not there is a connection between these Cenozoic features and the older Proterozoic structures is still debated (e.g., Caine et al., 2010; McCoy et al., 2005).

## 4 Running the Module

At the University of Colorado, we run this module as a three-part virtual field course. These include an introduction, Part I mapping, Part II additional analytical datasets, and Part III final reports and presentations. We begin with introductory lectures and warm-up activities such as ones that introduce concepts of field data uncertainty (e.g., Tikoff, 2020a), brief introductions to StraboSpot (e.g., Tikoff, 2020b; Walker et al., 2019), and mapping in relatively simple geologic terrain (e.g., Houghton et al., 2015; Houghton, 2020). These serve to (re-)familiarize students with key concepts and tools that they need for the mapping
exercise and to help overcome initial concerns or barriers to engagement (e.g., Stokes and Boyle, 2009; Orion and Hofstein, 1994). Additionally, warm-up assignments may illuminate individual student's strengths and weaknesses in order to assign complementary mapping partners for the following Part I exercise.

### 4.1 Part I Mapping

The mapping exercise is the foundation of this module. This component should take approximately 4 full days of fieldwork,
if run with an immersive design such as a traditional fieldcamp. However, when run during the semester, when students are also taking other courses, each "mapping day" is spread over a number of actual days. Students map in pairs to encourage teamwork, but also to simulate real fieldwork in which collaborative decision-making skills are required for planning and safety. To simulate a typical day of "field" mapping, we developed a "daily" mapping schedule. and describe it here in five steps: 1) introduction, 2) route planning and station selection, 3) mapping, 4) drop-in meetings, and 5) continued mapping.
Each step is briefly described below.

We typically begin each mapping day with a regularly scheduled class meeting. For the first day, this may be used to introduce the field area (see Geologic Background), goals for the mapping exercise, software, and logistics. Subsequent meetings are used to introduce more advanced geological concepts, discuss mapping strategies, and address student questions. Appendix A1 includes resources that may help introduce the field area through several different approaches. For example, the Powerpoint
slides introduce the main map units and some common lithological and structural features. The Google Earth Web project may be used to interactively explore satellite imagery, topography, station locations, and oriented photographs from the field area (Fig. 2). Additionally, the class may visit some key mapping stations together in order to demonstrate and practice collecting





"field data" from virtual outcrops. Some station descriptions include quite a lot of information, requiring students to recognize the most relevant data to collect for mapping, and some guidance from instructors may be helpful here.

The next step is for each student group to select their mapping stations for the day. We consider about 15 stations per day as reasonable given the terrain and common degree of complexity encountered at outcrops. Students can use the Google Earth Web project or other resources to plan their route using logistical considerations (e.g., safety, access, exposure) and geological reasoning. Before receiving station descriptions (virtual outcrops), students build a strategy for each "mapping day" by completing and submitting a station description request and justification form. The form requires students to list the selected stations, describe the route to access them, and calculate logistics such as round trip distance, time, and elevation change. This encourages students to use diverse rationale to choose their specific stations, as

**Figure 2.** View of field area from one of the vantage points provided in the Google Earth Web project.

they would in the field, instead of choosing random or consecutively-numbered stations. The form also includes specific prompts to explain how their planned strategy will test specific working hypotheses (based on previous mapping days), promoting iterative scientific reasoning that builds with each mapping day. By the end of the 4 mapping days, each group will likely have selected different combinations of 60 stations (out of 110) using varied strategies for their final maps, yet the class as a whole should have encountered most of the rocks, fabrics, and structures across the field area.

The third step is for students to begin mapping. This includes various tasks, such as setting up the basemap, extracting relevant outcrop data from station descriptions, and depicting those data on a map. We intend this module to use StraboSpot as a data management and mapping platform. Instructions for setting up the basemap with spot locations (Fig. 4) and populating them with data are described in Appendix A1 and supplemental materials. If accessibility prevents digital mapping, students can instead print and use a paper basemap (Appendix A1). However, StraboSpot offers some helpful digital mapping capabilities that are not achievable with paper methods, especially when including multiple attributes for each spot. The next task is to read, filter, and extract relevant outcrop data from each station description (Fig. 3). Strabospot allows each spot to host various types of data such as bedding, foliation, and lineation measurements for point spots, contact certainty or fault type for line spots, as well as rock type for polygon spots. Students are encouraged to include as much data as possible from the station descriptions into their map, as different types of data may be relevant for interpreting various geologic events. Satellite imagery and topography may also complement outcrop information. Students then use fundamental geologic principles to interpolate between mapping stations and across the entire field area, while developing multiple working hypotheses to explain map relations.



# GG24

**Lithology:** Quartzite.

**General structure:** This quartzite has a mylonitic penetrative foliation and stretching lineation defined by alignment of white mica and streaks of variably shaded quartz. Another prominent feature is a m-scale tight Z-fold (viewed down plunge) with a hinge that plunges shallowly northeast. Since the hinge plunges at a high angle to the stretching lineation, the asymmetry of the fold can be a high confidence (CL 5 on | scale from 1-5) shear sense indicator, implying **NW-side-up shear**. The outcrop is extensive enough to trace 4-5 meters of mylonitic quartzite across strike to the NW and then at least 10 meters of quartzite with much weaker but still steeply dipping and otherwise concordant (non-mylonitic) foliation.

**Measurements:** Mylonitic foliation strike, dip is 065,85. The lineation plunge -> trend is 79 -> 209. An independent measure of the axial surface of the m-scale fold is strike,dip 070,72 with a hinge plunge -> trend 30 -> 055.

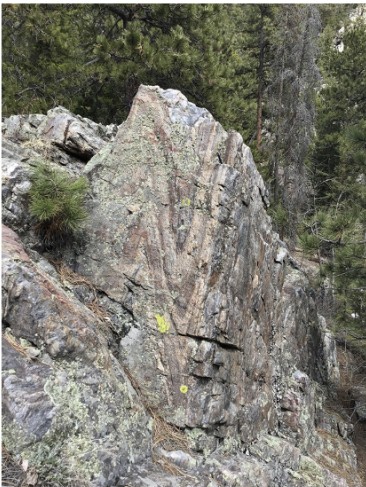

**Photo 1.** Left. Quartzite mylonite outcrop. Looking N. Photos 2 and 3 taken from near middle of outcrop. Field sketch below.

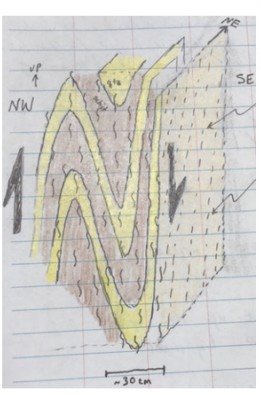

**Sketch.** Schematic field notebook illustration of the main features.

**Figure 3.** Part of a typical outcrop description.

The fourth step is for an instructor to remotely "drop-in" on student groups. In this regularly scheduled meeting, students screen share their current work (e.g., map) and observations with their group partner(s) and the instructor, describe their working hypotheses and plans to test them, ask questions and receive feedback. This mimics the concept of instructors episodically

joining a mapping team on the outcrops in the field. Two major advantages of these drop-in meetings are the abilities to interact during real-time mapping and decision-making and to digitally annotate on each other's shared screens and maps. At the end of this meeting, the instructor moves on to another group while leaving the students to continue mapping on their own.

Step five involves continued mapping with the rest of that days' stations, refining hypotheses, and planning the following day using the station request and justification form. Similar to in-person field mapping, we recommend emphasizing that it

may be difficult to revisit previous stations a second time. Therefore, students should complete as much mapping as possible by the end of that day. After this final step, the process repeats for the remaining three mapping days.



Concluding Part I, students should complete their geologic maps and other map products. Submitted maps may range from a simple screen-capture of the StraboSpot map to an exported and digitized publication quality map using GIS or graphics software. Optionally, students may complement their maps with written descriptions of map units and structures, as well as
cross-sections. In our module, instructors review and provide feedback on these initial deliverables at the end of Part I so that students can learn from and improve their work before the final submission in Part III.

In summary, Part I of the module provides an experience that addresses six out of the seven learning objectives in Table 1. These include designing a mapping strategy (Learning Objective 1), collecting enough information to plot on a map and guide map construction (Learning Objective 2), synthesizing that data to interpret map relations (Objective 3), developing
multiple working hypotheses based on their map data and certainty (Learning Objective 5), as well as presenting those data and interpretations to their partners and instructors via maps, cross sections, and station request forms (Learning Objective 6). In order to complete Part I efficiently, students also practice planning, time management, and teamwork skills (Learning Objective 7).

## 4.2 Part II Additional Datasets and Analyses

The mapping exercise in Part I is designed to be a stand alone part of the module, and some instructors may choose to only use that part. However, Parts II and III offer the opportunity for students to enrich their efforts by exploring what is possible by integrating additional laboratory tools. It is these latter two parts together that we feel makes the module a capstone experience. Because scientific projects rarely end when the mapping is complete, the students are challenged to see how samples and analytical data may commonly be collected and integrated with field observations to produce a more holistic understanding of
the geological history.

This part of the module introduces students to new perspectives on their previously mapped field area, both from additional analytical data and from descriptions of the same rocks and relationships through the eyes of previous workers
and their data. Students can work to relate their initial field-based findings with multiple advanced analytical datasets (Learning Objective 4). Exposure to a variety of types of uncertainty, and with a range of approaches to quantifying it, helps to address Learning Objective 5.

We also conduct this part of the module with students working in small groups, and in a similar style and time-frame as Part I. The datasets that are available to choose from include 1) Advanced field analysis, 2) Microstructural analysis, 3) Metamorphic petrology, 4) Metamorphic mon-
azite geochronology, 5) Igneous zircon geochronology, and

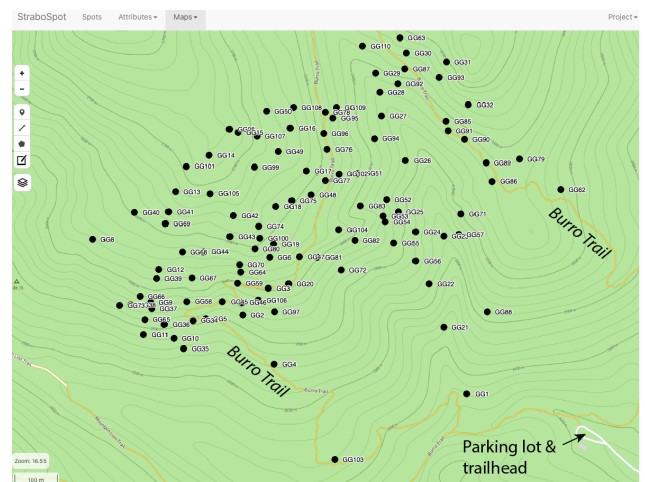

**Figure 4.** Map view from within a StraboSpot project where the initial shapefile with outcrop locations has been imported. Labels for parking lot and trails added in larger font for clarity.





6) Detrital zircon geochronology. A suggested sequence for

this part is as follows. First, student groups read the journal articles associated with their datasets and answer some guiding questions. Second, students review and summarize the fundamental background for the relevant techniques. Third, students compile their assigned supplemental data and conduct the relevant analyses. We recommend having student groups generate at

least one meaningful figure from their data and analyses, which generally involves recreating a plot from the assigned paper. This may allow students to better appreciate the meaning of the data, how it can be interpreted, and the range, sources, and magnitudes of uncertainty. Fourth, students interpret and synthesize the results with their prior hypotheses from the field-based mapping. Similar to Part I, we recommend periodic drop-in meetings with student groups throughout this process. Each dataset is described in more detail below, including the techniques and software to conduct analyses as well as suggested sources for

additional background information.

### 4.2.1 Advanced Field Structural Analysis

The first dataset includes all observations and data from the 110 stations, including measurements for bedding, foliations, mineral and stretching lineations, fold axial surfaces, fold hinges, enveloping surfaces, and shear sense observations. Students can analyze this data using free stereonet software such as Stereonet 11 (Allmendinger et al., 2013; Cardozo and Allmendinger,

2013), Stereonet Mobile, which pairs well with StraboSpot on tablets or smartphones (Allmendinger et al., 2017), or Orient (Vollmer, 2015). The suggested journal article is Shaw et al. (2002), a GSA field guide that includes a map, cross-section, and structural data presented in stereonets from an overlapping field area that includes the Idaho Springs-Ralston shear zone. Students can conduct similar analyses and produce similar plots to address questions such as 1) how do rock fabrics and structures generally relate from outcrop- to map-scales, 2) how does the pi-axis for S0-S1 data compare to outcrop-scale fold

hinges, 3) can the distribution of L1 stretching and mineral lineations be explained by F2 folding (e.g., use stereonets to unfold via techniques such as described in Ramsay and Huber (1987) and Duebendorfer (2003), and 4) how can the data be interpreted to represent multiple episodes of deformation. A final question appropriate for all dataset analyses is how does this additional level of analysis help them refine their initial interpretations?

### 4.2.2 Microstructural Analysis

The second dataset includes crystallographic orientations from a mylonitized quartzite from the Idaho Springs-Ralston shear zone (station GG2 in the map area), collected with electron backscatter diffraction (EBSD). This dataset, an optical photomicrograph and textural descriptions are from Ward et al. (2012). Students can generate a c-axis pole figure diagram using the MTEX toolbox (Bachmann et al., 2010) for MATLAB or the commercial Channel 5 software from Oxford Instruments. Tasks can include investigating 1) dislocation creep slip systems for quartz, 2) the temperature dependency of active slip systems in

quartz (Stipp et al., 2002), and 3) the use of c-axis patterns for determining shear sense. Reviews of the principles of crystal plasticity and the use of pole figures can be found in an introductory structural geology textbook such as Fossen (2016), and brief introductions to the EBSD technique are available in Fossen (2016, Ch.11 Box 4) and Swapp (2019). This analysis can help students relate microscopic to outcrop- and map-scale structures and observations, and it provides a different perspective





on some of the same questions that can be addressed with other group datasets (e.g., advanced field structural analysis and
metamorphic petrology).

### 4.2.3 Metamorphic Petrology

The third dataset includes a range of documented observations from Mccoy et al. (2005) on key metamorphic mineral assem-
blages (including kyanite, andalusite, and sillimanite), reaction textures, and relative timing of mineral growth with respect to
the multi-phase deformation history of the Coal Creek quartzite/schist in an overlapping study area. Secondary sources with
additional data are McCoy (2001, MSc thesis) and Ward et al. (2012). Students can use these data to 1) value the alumino-
silicate polymorphs as index minerals, 2) use reaction textures to build relative timing relationships among key minerals and
the main deformation events, and 3) semi-quantitatively construct Pressure-Temperature-time-Deformation (P-T-t-D) histories
for the metasedimentary and igneous rocks. A good online introduction to P-T-t paths and the basic principles and common
techniques behind their construction is Whitney (2020). This dataset can be used to help students outline the relative ages of
multiple tectonic events, understand the geodynamic implications of metamorphic histories, and to relate P-T paths to tectonic
settings, all of which can be further refined by groups who evaluate numerical dating techniques in the remaining datasets.

### 4.2.4 Monazite Geochronology

The fourth option is a U-Th-total Pb monazite geochronological dataset from mylonite and biotite gneiss in and near the Idaho
Springs-Ralston shear zone, also sourced from Mccoy et al. (2005) and described in somewhat greater detail by McCoy (2001,
MSc thesis). This is the dataset that most clearly documents the absolute timing of deformation and metamorphic events in the
study area, and students can use it to plot monazite date frequency distributions using Excel to distinguish multiple events. A
document on the basic theoretical background for U-Th-Pb geochronology and pros and cons of various analytical techniques
is provided, and it should be helpful for this dataset as well as Datasets 5 and 6. Concepts that students can explore include
1) the theoretical background for chemical vs isotopic dating of monazite, and 2) the potential for monazite growth to record
polyphase deformation and metamorphic events, including evidence in this dataset for both Paleoproterozoic (ca. 1.7 Ga) and
Mesoproterozoic (ca. 1.4 Ga) events.

### 4.2.5 Igneous Zircon Geochronology

The fifth dataset includes SHRIMP U-Pb zircon data from the Boulder Creek batholith (Premo and Fanning, 2000), which is
the granodiorite unit encountered in the map area. Students can 1) summarize the unique value of the U-Pb system having
two independent parent-daughter isotope sequences, 2) explain the concept of concordia and plot some of the Boulder Creek
data using the provided basic Excel template for a concordia diagram, 3) evaluate how Premo and Fanning (2000) derive their
preferred age for the Boulder creek batholith and how that provides constraints on the timing of deformation events in the map
area, and 4) understand the meaning of zircon inheritance in magmatic rocks, including the implications of Archean inherited
zircon in the Boulder Creek samples. In addition to the U-Th-Pb basics document provided, students can also be steered towards





additional introductory tools for in situ geochronology such as the ZirChron app (Schmitz and Viskupic, 2014). Importantly, this dataset holds a clue to interpreting the contact between the granodiorite and adjacent Coal Creek quartzite/schist, which may be unresolved by Part I mapping alone. Another clue resides in the final dataset.

### 4.2.6 Detrital Zircon Geochronology

The sixth dataset includes detrital zircon U-Pb LA-ICP-MS data from several Proterozoic quartzite occurrences in Colorado
including the Coal Creek quartzite/schist sequence (Jones and Thrane, 2012). Students can plot this data as age frequency histograms using Excel and on concordia diagrams using the provided Excel template. Some concepts to steer students towards could include how detrital zircons 1) inform on sedimentary provenance and 2) provide constraints on the depositional age of (meta)sedimentary rocks. Importantly, this dataset also provides another clue to the relative age of the Coal Creek quartzite/schist versus the Boulder Creek granodiorite. The latter two datasets are essential to deciphering the tectonic sequence
of initial batholith emplacement at depth, rapid exhumation and sedimentary recycling, and subsequent burial, deformation, and metamorphism as revealed with the other datasets.

### 4.3 Part III Written Reports and Oral Presentations

This module culminates with students tying each part of their work into a holistic presentation. We end our course with student groups orally presenting their work to the class and writing a report. During presentations, groups emphasize their
unique datasets and analytical techniques, demonstrating how their results and interpretations contribute to the entire class's collaborative effort. The short written report includes maps, cross-sections, and figures from Parts I and II. The presentations stimulate students to think critically about the scientific motivation for conducting research, and the implications of their final results and interpretations. It is also likely that students' initial map interpretations are modified after analyzing their Part II supplemental datasets and learning from other groups' presentations. Therefore, the students gain a better understanding of how
the scientific process continues to build on previous work. Part III teaches how to complete a scientific project with discipline-specific communication skills (Learning Objective 6), effectively helping to train them for academic research, conferences, and/or applied science.

## 5 Assessment of Student Learning

Course assessment can take many forms, and the most effective ones for field modules are tailored to the specific field ex-
periences (e.g., Pyle, 2009). We suggest several mechanisms that can provide a combination of formative and summative assessments in this module. First, in our version of the module, students are asked to fill out pre- and post-course assessment forms. These forms start by prompting students to numerically rate their confidence level (1-low to 5-high) with each of the 7 learning objectives identified in Table 1 and with the utility of the 6 analytical techniques outlined in Part II. Students also explain, in short written form, their current understanding of how to use these analytical data and techniques. These forms
serve two purposes. One is to assess learning via self-reported data from the beginning to end of the course. Another is to



inform instructors on student strengths and weaknesses at the beginning of the course, in order to help pair students with complementary skills into working groups, assign suitable analytical datasets and techniques, and to modify teaching strategies that cater to students' individual needs. A second mechanism is the Station Request and Justification Forms that students use for each of their 4 "mapping days." Instructors can use these data to track and provide ongoing feedback on student mapping

progress, how well students develop their understanding of the map relations over time, and how comfortable students become with hypothesis testing. The third mechanism is the instructor "drop-in" meetings, which provide individualized guidance to students in real time. These meetings allow instructors to evaluate progress and assess students' processes of data collection and interpretation via live interaction, instead of just through the final products. Fourth, students periodically submit draft deliverables (e.g., exported "field notebooks" from StraboSpot, maps, and written reports) throughout the module. Instructors can

assess the quality and content of these draft products and return constructive feedback and expectations to students before the final products are due. Finally, summative assessment can be addressed through the suite of graded assignments.

## 6 Advantages and Disadvantages of Virtual Field Modules

The disadvantages of a virtual field course are some of the most obvious. They include the absence of hands-on field data collection, the inability to see outcrops in person, the lack of face-to-face student and instructor interaction, as well as the

missing appeal and unique learning benefits of working outdoors in a natural environment (e.g., Mogk and Goodwin, 2012).

While a virtual course cannot replace the actual field experience, modules like the one described here can successfully address generalized Learning Objectives (Table 1) and maintain, or potentially improve, certain aspects of a field-based capstone experience. One benefit is how this and other virtual courses can use technology to resolve conceptual difficulties that students commonly encounter in field experiences, such as interpreting 3-D structures from 2-D data, linking geologic features across

scales, and understanding the temporal or evolutionary relationships between geologic events (Whitmeyer et al., 2009a). For example, the Google Earth Web project provides a spectacular resource for initial reconnaissance and route planning. It can also be used throughout the mapping exercise to identify geological influences on outcrop exposure or vegetation, as displayed with satellite imagery, or to visualize 3-D structures by obliquely-viewing digitized 2-D map features that are draped over the 3-D topography. Additionally, the ability to selectively display map features in StraboSpot, as well as stereonet software, allow

students to relate cm-scale folds at an outcrop to km-scale folds across the entire mapping area. Some of the analytical datasets extend this connection to microscopic scales (e.g., quartz microstructure and sub-mm zircon and monazite geochronological targets). While many of these technologies are increasingly accessible for actual field work, virtual modules remove some of the hurdles of working with them offline.

Another advantage of this virtual module is the simultaneous online communication, which facilitates some new forms

of interaction and comprehension that were not possible before. For example, screen sharing and live digital annotations during group mapping and drop-ins provide potentially powerful learning resources. Students can draw their hypothesized map relations or structures, instructors and partners can view the students' mapping ideas in real-time, while students and instructors alike can contribute with live interactive feedback. Importantly, digital annotation features have the "undo button",



which promotes experimental mapping and hypothesis testing, without risk of ruining the map (e.g., permanent marking on
paper maps). Through informal feedback, students have expressed that they especially value the drop-in meetings.

Finally, this experience fosters a more accessible and inclusive learning environment for all students. Notably, it only required
access to a computer and internet, in contrast to other field courses with prerequisites of physical ability and comfort in the
outdoors. Several of the design features recommended by Stokes et al. (2019) for making fieldwork accessible are achieved
with the structure of this module, including a flexible pace, focus on collaborative learning, and emphasis on the academics of
collecting and synthesizing field-based data as opposed to physical rigor.

## 7   Conclusions

We have described a module for remote delivery that encompasses many of the basic requirements of an advanced field
exercise, including designing a field mapping strategy, collecting and processing field observational data, and synthesizing
and communicating the results in a variety of ways. As a whole, the module covers concepts of basic mapping practices,
3-dimensional structure, polyphase deformation, relative vs. absolute timing, geochronology, microstructure, P-T histories,
tectonics, and uncertainties in both field and laboratory data. Incorporating computing technology, including new field data
collection tools, virtual globes, and analytical software, prepares students for careers in industry or academia, and offers a wide
range of pedagogical benefits (e.g., Whitmeyer et al., 2009a).

The module encourages students to explore the important links between field and laboratory work, which may appeal to a
wider range of students than those who are generally attracted to one or the other mode of scientific investigation. Modifications
to incorporate additional visualization tools may be relatively straightforward, such as converting structure data into 3-D
symbology in Google Earth (Blenkinsop, 2012) or draping geologic maps over digital elevation models. Similar approaches
could be adapted to other localities where a range of analytical datasets are available.

*Data availability.*   All materials associated with this teaching module are accessible through the Supplement and are described below in the
Appendix. These materials, including higher resolution outcrop photos in the station descriptions, are also available at the NAGT Teaching
with Online Field Activities website (Mahan and Frothingham, 2021).

## Appendix A:   Description of Materials

### A1   Part I: Mapping

**Station Descriptions and Data** *GGstationdescriptions.zip* The zipped folder contains station descriptions and data (e.g., lithol-
ogy, outcrop and structural descriptions, photos, sketches, and strike/dip and/or trend/plunge measurements). Some stations do
not contain all of these components. Each file is a Word document (.docx). Higher resolution versions of any of the photos are
available upon request from Mahan.





**Google Earth Web "Golden Gate Mapping Project"** Google Earth Web This project includes the field station locations as well as 9 View points containing general photo perspectives of the field area. Click on the View points in the listing on the left,

and the screen will zoom to the 3-D perspective from which the photo was taken. Alternatively, one can save a KML file of this same project from within GEW, which can be viewed in Google Earth Pro, although the photos at the View points will not be available. There is a trail system in the field area but they are not marked in Google Earth Web at the time of this writing. However, they are available with the topographic basemap provided in StraboSpot (Fig. 4)

**Station Locations Shapefile** *GGstationlocations.zip* The zipped Shapefile contains all 110 station locations that can be up-

loaded to StraboSpot. Use this to pre-populate the 110 station locations as Spots (the file contains locations only). Students can then add data to the spots, and turn off the display of the ones that they do not use.

**Instructions for StraboSpot** *InstructionsStraboSpot.docx* Word document listing step-by-step instructions for getting started with StraboSpot, including how to load the above Shapefile, and some tips for setting up and conducting the mapping project in Strabo.

**Station Description Request Form** *StationRequestForm.docx* A blank station description request form as a Word document - students (or pair of students as mapping partners) fill out this form in order to justify their strategy for the next day's mapping.

**Basemap Template** *GGBasemap.pptx* Basemap template in PowerPoint for optional construction of final map. The working geologic map is built in StraboSpot, and with some effort a final map can be made to look quite nice and can be screen-captured for grading. However, this file is provided if a more polished final map is desired.

**Map unit Introductory Slides** *IntroGGmapunits.pptx* Powerpoint file with several slides that can optionally be used to introduce the rock types in the map area.

**Assessment form** *Preassessment.docx* A pre-course assessment form for self-reported data as a Word document. This can be tailored to specific teaching and learning interests, and it can also be modified for use as a paired post-course assessment.

**Summary for Intructors** *GGmapsstereonetsummaries.pdf* A pdf document for instructors with screenshots of draft geologic

maps (several versions showing different Spot datasets) made from all stations, and summary stereonets of the basic structural components.

## A2   Part II: Analytical Datasets

**Introduction to Part II**: *Introductionanalyticaldatasets.docx* is a word document with brief description of 6 optional datasets, to aid students in the choice of dataset with which they will work. *Introanalyticaldatasets.pptx* is a set of Powerpoint slides

that can be used for the same purpose.

**Dataset1**: *Dataset1AdvancedStructure.docx* Document to be provided to students (or small groups of students) after they have made their choice. This document provides full references for data, a list of goals, recommended figures to create, and a list of specific questions or topics to address in the final report and presentation. *GoldenGatefielddata.xlsx* An excel file with the same text for each station that is found in the individual station descriptions, and the structural measurements. This is provided for

use as the advanced field structural dataset, and it can be used to generate subsets of data for exporting to stereonet construction software.





**Dataset2**: *Dataset2Microstructure.docx* Document to be provided to students (or small groups of students) after they have made their choice. This document provides full references for data, a list of goals, recommended figures to create, and a list of specific questions or topics to address in the final report and presentation. *Wardetal2012ISRq.ctf* A Channel Text File

for sample ISR-q from Ward et al. (2012), which was collected at station GG2 in this module. This can be used in MTEX (Bachmann et al., 2010) or in Oxford Instruments' Channel 5 suite of software to explore EBSD data, including generating a quartz c-axis pole figure. The thin section from which the EBSD data was collected was prepared such that the viewer is facing NE and looking at the plane perpendicular to the mylonitic foliation and parallel to the stretching lineation. The resulting quartz c-axis girdle is asymmetric and indicative of NW-side-up sense of shear.

**Dataset3**: *Dataset3MetamorphicPetrology.docx* Document to be provided to students (or small groups of students) after they have made their choice. This document provides full references for data, a list of goals, recommended figures to create, and a list of specific questions or topics to address in the final report and presentation.

**Dataset4**: *Dataset4MetamorphicMonazite.docx* Document to be provided to students (or small groups of students) after they have made their choice. This document provides full references for data, a list of goals, recommended figures to create, and a

list of specific questions or topics to address in the final report and presentation. *McCoy01mnzISRsz.xlsx* An excel spreadsheet of raw U-Th-totalPb and Y data from the Idaho Springs-Ralston shear zone from McCoy (2001).

**Dataset5**: *Dataset5IgneousZircon.docx* Document to be provided to students (or small groups of students) after they have made their choice. This document provides full references for data, a list of goals, recommended figures to create, and a list of specific questions or topics to address in the final report and presentation. *PremoandFanning2000data.xlsx* An excel

spreadsheet of U-Pb SHRIMP igneous zircon data for the Boulder Creek Granodorite from Premo and Fanning (2000).

**Dataset6**: *Dataset6DetritalZircon.docx* Document to be provided to students (or small groups of students) after they have made their choice. This document provides full references for data, a list of goals, recommended figures to create, and a list of specific questions or topics to address in the final report and presentation. *JonesThrane12Appendix1DZdata.xlsx* Excel spreadsheet of U-Pb LA-ICPMS detrital zircon data of quartzite from Jones and Thrane (2012).

**Geochronology Review**: *U-Th-Pb Basics.pdf* An introductory handout describing the basics of radioactive decay, U-Pb decay chains, the age equation, and analytical methods for U-Pb zircon analysis and U-Th-Pb$_{Total}$ analysis of monazite.

**U-Pb concordia spreadsheet**: *Concordiadiagram.xlsx* is an Excel spreadsheet formatted for plotting U-Pb data on a concordia diagram. It is not formatted to include error ellipses. Students can copy select data from a source file and make new concordia diagrams with this spreadsheet.

## A3 Part III: Reports

**Written and Oral Reports**: *Intructionsforfinalreport.docx* A Word document providing detailed recommendations for the organization and what to include in the draft and final written reports and a possible grading rubric.





*Author contributions.* K.Mahan conceived of the idea for the module and developed the outcrop descriptions and field structural dataset, the Google Earth Web project, compiled the analytical datasets, the basic module structure, and he helped write this manuscript. M. Frothingham

was an invaluable graduate teaching assistant for the first two CU Boulder versions of the course. He developed most of the activities employing computer software, visualization and interactive pedagogical tools, contributed important components to the module structure, and helped write this manuscript. E. Alexander provided the U-Th-Pb basics document and helped write this manuscript.

*Competing interests.* The authors declare that they have no conflicts of interest.

*Acknowledgements.* We thank the students who took the first two courses that used this module at CU Boulder. They provided extremely

valuable feedback on the content and its effectiveness. K. Hannula used an early version of this module for teaching her students at a separate institution and she provided very helpful suggestions for improvement. L. Arthurs provided valuable advice on developing assessment tools.



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
