# Peer review of "Virtual mapping and analytical data integration: A teaching module using Precambrian crystalline basement in Colorado's Front Range (USA)"

_Geoscience Communication, 2021_

## Author Response (AR1)

**Mahan et al. – authors' response**

**Reviewer 1 comments:**
The manuscript rating explanation:

**Scientific significance: 3** – There is nothing state-of-the-art in this article, but it does provide a nice case study of data integration using a newer platform. The lack of any real results or reflection also makes it difficult to assess whether this method is useful.

**Scientific quality: 3 –** The method is well-described and refers back to learning objectives in a good way. A little more balance with respect to chapter 6 would be useful to others in the same field. The use of the term "virtual outcrops" needs to be addressed/clarified.

**Presentation quality: 2 –** The article is clear and well-presented. I would like to see a figure showing the finished product of one of these maps.

General Commentsu

This article provides an overview of a digital mapping activity designed out of necessity, due to the COVID-19 pandemic, to enable students to continue developing their field-mapping skills. The article is well-written and structured with good reference to the learning objectives defined in Table 1.

While the paper provides an interesting case study, I find it very difficult to assess how effective the teaching method actually was. The authors mention questionnaires sent to students but then provide no results, which I feel would be fascinating. I feel the advantages/disadvantages is more of a sales pitch (after the first paragraph) rather than providing useful information to other geoscience lecturers. It would also be useful to see an example of a completed map project here.

In my opinion, the main weakness of the paper is the overuse of the term "virtual", when in reality the only virtual geology here is in the use of Google Earth projects. The "virtual outcrops" are really Microsoft Word documents with field observations, interpretations and measurements provided to the students. A "virtual outcrop model" (e.g., 3D photogrammetry, lidar etc.) will allow students to perform their own observations and measurements (the most important skill to all geologists). However, I think this weakness is also the biggest strength; it demonstrates how good geological record keeping and data management can be, quite quickly, integrated using more modern software to great effect and is something that should be highlighted.

While I am critical of the use of "virtual" in individual elements, the overall package and delivery method is virtual, and it fits well with the Special Issue theme. I also think the authors should be commended for providing the course material to enable others to virtually take this course or gain more specific insights.

**Specific Comments**

Line 31: A specific reference with respect to the "improved mapping tools" would be useful for the reader.

Line 51: As mentioned in general comments: these are not virtual outcrops, they are field observations, measurements and interpretations from the outcrops or "stations".

Line 82-84: I still don't quite understand what StraboSpot is from this description, and it is clearly quite an important tool for this article based on wordcount. I know the Walker et al. article addresses this, but it would be useful to know if it is a mapping platform, data management and storage, or a field tool? (or all of these?).  A figure showing an example map would be useful.

Line 144: What is the resolution of the "Google Earth Web project"?  What is its purpose? Do the students carry out any larger scale analyses on it or is it simply for planning and visualizing the station locations?

Line 164: A little more information on why you prefer StraboSpot over other alternatives would be interesting. But this would probably be resolved with my comments re: line 82-84.

Line 167: Please clarify a little. This sounds like students are simply asked to copy as much data from a word file into a different piece of software? That seems to contradict the "filter" data part 2 lines prior.

Line 220: What do you mean by "recreating" a plot?

Lines 233-235: Explain the abbreviations/concept for structure types and their generation (e.g. F2 is folding $2^{nd}$ gen.).

Chapter 4.2.3 - 4.2.6:

Although these are clearly good datasets for the students, there is a lot of superfluous information with respect to the context of this article. I also struggle to see the integration with mapping with some datasets – this would probably be resolved by simplifying and/or showing a completed map.

Chapter 5

Do you have any results from these questionnaires?

Chapter 6

As mentioned in general comments: I think this chapter would be nice with some of the results of student feedback and a bit more balanced reflection.

I am interested in how this differs to how the module was run prior to COVID when fieldwork could occur. Also a little about the future of the course beyond COVID would be interesting, have learnings helped to develop the field-based? Do you plan on turning any of the stations into 3D models?

**Technical Corrections**

Line 17: remove hyphen from "on-line"

Line 314: I guess "4" should be "four".

**AUTHORS' RESPONSE TO REVIEWER 1 COMMENTS:**

From online discussion response on 6/16/21:

*Thank you very much, Reviewer 1, for your helpful review! We offer responses to the main comments below.*

*Reviewer 1 appears to have two main concerns in their general comments. The first is that we did not include response data from our pre- and post-course questionaires. While it is true that we have ran this module twice now (July 2020 and Spring 2021) and we used the questionnaires in both courses, we cannot legally publish the results because we were not certified by CU's Institutional Review Board to conduct research on human subjects. However, we will look for ways to share more generalized insight from these responses and will plan to include this in our revised manuscript. One item that is already shared in the current manuscript is how much the students valued the "drop-in" meetings with individual mapping partners. Another point that we can make in the revised manuscript, a sort of "lesson learned", is that we did not use the current version of the "Station description request and justification" form in our first running of the module – specifically, the prompts for hypothesis-testing were added in between our two courses. We found that the students needed to be held more specifically accountable for justifying why they wanted to "map" in a particular area on each successive day. The modified, and now current, version of this forms addresses this initial shortcoming.*

**Action taken to revise:** We added two sentences in section 5 – Assessment of Student Learning – to acknowledge this. Page 14, lines 340-342.

*Another point that we can make in this context, and this also addresses a later question by the reviewer, is that we do intend to incorporate aspects of the dataset integration into the in-person field-based version of this course.*

**Action taken to revise:** We added a statement to the last paragraph of the conclusion section (section 7). Page 15, lines 388-390.

*The second concern is with our use of the term "virtual" for individual station descriptions. We have no objection to the reviewer's comment here, and we acknowledge the distinction that the reviewer makes with examples of what are probably better examples of "virtual" outcrops.* ==We will simply refer to the station descriptions as "station descriptions" in the revised manuscript.== *And we also agree with the reviewer that the overall module is appropriately considered "virtual" and so we will opt to keep that term in the manuscript title. Also, in reference to a later question from the viewer, we do not have immediate plans to convert some of these stations into what others might consider more conventional virtual outcrops, with photogrammetry for example. This is beyond the scope of our intent.*

**Action taken to revise:** There were four places where we originally used this phrase - sections 2.1, 2.2, and 4.1. All have either been removed or changed to "station descriptions." Page 2, line 54. Page 3, line 59. Page 5, line 74. Page 8, line 165.

*Reviewer 1 also made some specific comments by line item. These are all very helpful and we will address them in our revised manuscript. We highlight a subset with responses below.*

*Line 82-84: Strabospot is a combination of a mapping platform, data management and storage tool, and field data collection tool.* ==We will clarify that in the revised text.== *A figure with an example map is also requested – this is already included in the Supplemental materials – it is possible that the reviewer missed this (it is understandable since there are lot of supplemental files!)*

**Action taken to revise:** We added a clarifying statement in the Technology requirements section 2.3. Page 6, lines 92-93.

*Line 144: In reference to the Google Earth Web project, this is for planning and better visualization of the stations only – it is not required. While the satellite imagery resolution appears to be a little better than what is provided by StraboSpot, the main advantage is the extra 3-D visualization provided by Google Earth draping the imagery over a digital elevation model, which is not available in StraboSpot.* ==We will try to make this more clear in the revised manuscript.==

**Action taken to revise:** We added clarifying statements in the Technology requirements section 2.3. Pages 5-6, lines 89-97.

*Line 167: Request is to clarify what students are requested to do for each station.* ==This is a really good point that we will clarify in the revised text.== *We initially encourage students to copy and paste the text description of the outcrop information in its entirety into the "Notes" section of a Spot in Strabo – simply because it streamlines the process of going back to review ones notes later. But then the "filtering" process is where the students must decide which aspects of this description, or what specific data, are most important for mapping and working out field relationships. For example, in some outcrops, multiple orientation measurements may be provided, but perhaps only one that could be considered "representative" is necessary for entering into the specific orientation section of the Spot so that a strike and dip symbol can be displayed on the map. Or, as another example, some outcrops display conglomeratic horizons*

*in the quartzite – this may not seem important to a student initially, but later they might decide that they want to go back to the descriptions and "tag" those outcrops with conglomeratic horizons for display on the map. It is quicker to do that if the full text description is already available in the Spot entry because it avoids the necessity of going back to the original Word files.*

**Action taken to revise:** We added several clarifying sentences to the original 4th paragraph (now 5th) in section 4.1 for clarity. Pages 8-9, lines 182-192.

*Line 220: Reviewer asks what is meant by "recreating" a plot. ==This could also benefit from some more clarification, which we will do.== In most cases, "recreating" is not the best word; and we refer the reviewer and other readers to the phrase "meaningful figure" in the first half of the sentence in the current manuscript. The most straightforward form of this is that students who are working with one of the U-Pb geochronology datasets might be asked to pick a subset of the published data and plot it on their own version of a Concordia diagram – the result can then be compared with one of the diagrams from the publication, and the idea is that they gain a deeper appreciation for what these data mean, associated uncertainties, and how to read such diagrams. Another example is that for the advanced structure dataset, students might be directed to generate stereonets of structural data "collected" from this field area as part of this module – these data are not published elsewhere so "recreating" is definitely not the right word. But instead, there are published stereonets from overlapping areas to which the diagrams created by the students can be compared. The subsections in Section 4 on the individual datasets give some more detailed suggestions for what diagrams students can be asked to generate.*

**Action taken to revise:** We replaced "recreating a plot" with "selecting and plotting a subset of data." The rest of the clarity was already there, but using the word "recreate" was misleading. Page 11, lines 244-245.

*Line 233-235: Explain use of abbreviations for structure types (e.g., F2). Good point. ==We can explain this briefly, and provide a reference. But it is also important to point out that this notation is not imposed by any aspect of the module materials – the outcrop descriptions do not use this notation. Instead, this notation is to be determined by students with guidance from instructors.==*

**Action taken to revise:** We added a reference to section 13.2 from Fossen 2016 structural geology textbook, which explains this notation, to the 5th paragraph in section 4.1. Page 8, line 187.

*Chapter 6: In reference to comparison with course before COVID and future of the course, this is an interesting point – prior to the pandemic, we did not incorporate the analytical datasets. I had not thought to do it before. But it has been such a success, that I do plan to incorporate these datasets into the in-person field-based version of the course in the future. ==We can add this reflection to the revised text.==*

**Action taken to revise:** As noted earlier, we added a statement to the last paragraph of the conclusion section (section 7). Page 15, lines 388-390.

*There are other minor comments and corrections for which we thank the reviewer.*

**Action taken to revise:** Additional minor changes throughout the document help address some of these points. See the tracked changed pdf for details of all changes.

**Reviewer 2 comments:**
General comments:

Overall, the manuscript is clearly written and does a nice job of outlining and documenting the authors' 'ready-to-go' virtual field module. I have worked through several parts of the module and am impressed by the amount of material the authors have compiled and included in the exercise. In the manuscript, I appreciate that the authors clearly state the learning objectives. In the module, the questions at the end of each exercise in Part II do a nice job of getting students to think about how different types of data (e.g. mapping and lab) can be integrated. The questions prompting students to address sources of uncertainty are especially valuable in promoting higher-order thinking.

I think the manuscript could be improved by addressing the following:

First, I manuscript would benefit from a section discussing how effective the module is in achieving some of the discipline-oriented learning objectives (e.g. 1-4 from Table 1). My understanding is that a version of this exercise has been used several times, which means the authors may have some insight into how an exercise like this translates to students who then work in the field? A further discussion of student survey results would also be interesting and provide useful information for instructors who are considering this exercise in their curriculum.

Second, the text would be clearer with slight reorganization or addition of a 'module outline' or something along those lines. Excluding two sentences in the paragraph at the end of the introduction (lines 35-38), the authors provide no indication of the module structure or content until Section 4. In my opinion, an outline of the project early in the text would serve the reader well. For example, throughout Section 2, the authors refer to different aspects of the module without having explained the overall module structure (line 74 refers to a 'Part III', which is the first mention of this as far as I can tell. Also, Section 2.1 – Learning Objectives – refers to specific parts of the module which have not been clearly explained).

In terms of the module, I have a couple of comments and/or points of discussion:

From working through parts of the module, in my opinion, the main strength comes from the integration of different data types (i.e., Part II). My main issue is not so much a critique of this project as it is a perceived shortcoming of attempting to have students 'map' virtually – and this is coming from the perspective of someone who supports the use of virtual exercises in the classroom. I suspect that in an exercise like this where students pick stations that already have

units identified, measurements, and interpretations, they potentially miss out on developing many of these field skills that are necessary to produce a map. Admittedly, these skills are not listed by the authors as learning objectives.

Regardless, I am curious to hear the author's thoughts on how effective Part I of the project was in teaching students how to map in the field (related to my first comment above). To the credit of the authors, they do add the exercise of picking out new stations on a daily basis and ask students to provide justification of their choices. While this certainly helps encourage students to think critically and strategize, it does not capture the real-time decision making that is an important field skill. I wonder if releasing stations to students multiple times throughout the day and requiring them to choose stations within a restricted geographic region relative to their most recent selection could be an alternative way to run this exercise that would help students develop these skills?

**Specific comments:**

Many of the station files have interpretations already included. Maybe it would be beneficial to student learning to prompt this sort of interpretation from the student (e.g., show the shear sense indicators and ask students to determine the sense of shear)?

Section 2 - As mentioned in the general comments section, at this point in the manuscript the reader has only a rough idea of what the module comprises. So the discussion of how learning outcomes are achieved, what materials are provided, etc. is difficult to follow.

74 – What is 'part III'? Also, you never properly defined parts I and II.

80 – What is the 'google earth web project'?

307 – Personally, I would be interested in seeing a summary of the pre- and post-course assessment forms.

**Technical corrections:**

16 – COVID-19 in abstract, Covid-19 in introduction

17 – 'on-line' should be 'online'

41 – no hyphen in 'pre-requisite'

79 – should be StraboSpot

81 – '3-d' here, but '3-D' in the latter sections of the manuscript. '3-dimensional' is used in places.

85 – I am not sure if 'set-up' should be hyphenated?

124 – is 'fieldwork' the appropriate term?

125 – The first 'when' is unnecessary.

229,285 – Should be 'these data' (may be more instances throughout)

240 – Personally I would prefer 'mylonitic quartzite'

Dataset2microstructure – questions to answer #2. 'quartzite' misspelled

Dataset6detritalzircon – questions to answer #3 is vague and may be better framed as a question.

**AUTHORS' RESPONSE TO REVIEWER 2 COMMENTS:**

From online discussion response on 6/17/21:

*Thank you very much, Reviewer 2, for your helpful review! We offer responses to the main comments below.*

*Reviewer 2 expresses four main concerns in their general comments. The first is the desire for some more specific reflection on how effective the module is with respect to discipline-oriented learning objectives. And that we did not include response data from our pre- and post-course questionaires. As we responded to a similar comment from Reviewer 1, while it is true that we have ran this module twice now (July 2020 and Spring 2021) and we used the questionnaires in both courses, we cannot legally publish the results because we were not certified by CU's Institutional Review Board to conduct research on human subjects. However, we will look for ways to share more generalized insight from these responses and will plan to include this in our revised manuscript. One item that is already shared in the current manuscript is how much the students valued the "drop-in" meetings with individual mapping partners. Another point that we can make in the revised manuscript, a sort of "lesson learned", is that we did not use the current version of the "Station description request and justification" form in our first running of the module – specifically, the prompts for hypothesis-testing were added in between our two courses. We found that the students needed to be held more specifically accountable for justifying why they wanted to "map" in a particular area on each successive day. The modified, and now current, version of this forms addresses this initial shortcoming. ==We will add these reflections to the revised manuscript.== Unfortunately, we are not aware of students who took our course and then conducted field work, and so we have no specific insight from that potential perspective. It is still probably too early to know.*

**Action taken to revise:** We added two sentences in section 5 – Assessment of Student Learning – to acknowledge this. Page 14, lines 340-342.

*The second point is that the reviewer suggests adding a module outline early in the manuscript. We acknowledge this good point, and will plan to add an explicit outline of the module early in the revised text.*

**Action taken to revise:** We modified some minor existing text in the Intro to include "Part I, II, III" so the outline is now there in words. We also added an additional figure – now Figure 1 – which is a simple flow chart illustrating components of the module. Page 2, lines 35-39. The new figure is on Page 3.

*The third point is that the reviewer suspects releasing data once each day does not capture the real-time decision making that is an important field skill. We are in complete agreement here - this is where a virtual exercise like this definitely falls short (most will, we suspect). The module is not designed to help students develop basic field data collection skills, for example, initially identifying relict bedding or different generations of foliations or folds, or how to measure the orientations of these features. Instead, we have to assume that these basic skills are in place and if so, here's an opportunity to take that data forward and work with it at the next level of relating outcrop-scale features to map-scale features. And also as the reviewer points out, it is an opportunity to integrate the field-based data with other types of data.*

**Action taken to revise:** We added a clarifying statement acknowledging this in the last paragraph of the Introduction section 1. Page 2, lines 40-42.

*The fourth point is that the reviewer wonders if a modified version of data release (several times per day instead of once per day) or modifying the station descriptions to take some of the initial field interpretation such as prompting students to interpret shear sense rather than having it provided would be advantageous – our response is that our description of what we did are suggestions only, whereas other potential instructors of course have the freedom to modify this. We will make this point in the revised text.*

**Action taken to revise:** We added a clarifying statement addressing this in the Materials provided section 2.2. Page 4, lines 71-73.

*Reviewer 2 also made some specific comments by line item. These are all very helpful and we will address them in our revised manuscript. We highlight one with a response below.*

**Action taken to revise:** Additional minor changes throughout the document help address some of these points. See the tracked changed pdf for details of all changes.

*Line 80: What is the "google earth web project?" Reviewer 1 had a similar question, so obviously we did not make this clear enough initially, and we will clarify in the revised manuscript. The GEW project is for planning and better visualization of the stations only – it is not required. While the Google Earth satellite imagery resolution appears to be a little better than what is provided by StraboSpot, the main advantage is the extra 3-D visualization provided by Google Earth draping the imagery over a digital elevation model, which is not available in StraboSpot.*

**Action taken to revise:** We added clarifying statements in the Technology requirements section 2.3. Pages 5-6, lines 89-97.